# Validation and Data-Integration of Yeast-Based Assays for Functional Classification of BRCA1 Missense Variants

**DOI:** 10.3390/ijms23074049

**Published:** 2022-04-06

**Authors:** Francesca Bellè, Alberto Mercatanti, Samuele Lodovichi, Caterina Congregati, Chiara Guglielmi, Mariella Tancredi, Maria Adelaide Caligo, Tiziana Cervelli, Alvaro Galli

**Affiliations:** 1Yeast Genetics and Genomics, Laboratory of Functional Genetics and Genomics, Institute of Clinical Physiology, CNR via Moruzzi 1, 56125 Pisa, Italy; f.belle01@hotmail.com (F.B.); alberto.mercatanti@ifc.cnr.it (A.M.); 87samuele@gmail.com (S.L.); tizicerv@ifc.cnr.it (T.C.); 2Division of Internal Medicine, University Hospital of Pisa, 56125 Pisa, Italy; c.congregati@ao-pisa.toscana.it; 3Molecular Genetics Unit, Department of Laboratory Medicine, University Hospital of Pisa, 56125 Pisa, Italy; chiara.guglielmi.cg@gmail.com (C.G.); mariella.tancredi@ao-pisa.toscana.it (M.T.)

**Keywords:** BRCA1, yeast-based functional assays, variant of uncertain significance, variant classification, homologous recombination, gene reversion, machine learning

## Abstract

Germline mutations in the BRCA1 gene have been reported to increase the lifetime risk of developing breast and/or ovarian cancer (BOC). By new sequencing technologies, numerous variants of uncertain significance (VUS) are identified. It is mandatory to develop new tools to evaluate their functional impact and pathogenicity. As the expression of pathogenic BRCA1 variants in *Saccharomyces cerevisiae* increases the frequency of intra- and inter-chromosomal homologous recombination (HR), and gene reversion (GR), we validated the two HR and the GR assays by testing 23 benign and 23 pathogenic variants and compared the results with those that were obtained in the small colony phenotype (SCP) assay, an additional yeast-based assay, that was validated previously. We demonstrated that they scored high accuracy, sensitivity, and sensibility. By using a classifier that was based on majority of voting, we have integrated data from HR, GR, and SCP assays and developed a reliable method, named yBRCA1, with high sensitivity to obtain an accurate VUS functional classification (benign or pathogenic). The classification of BRCA1 variants, important for assessing the risk of developing BOC, is often difficult to establish with genetic methods because they occur rarely in the population. This study provides a new tool to get insights on the functional impact of the BRCA1 variants.

## 1. Introduction

Germline pathogenic variants in the tumor suppressor gene BRCA1 strongly predispose one to hereditary breast and ovarian cancer (HBOC). BRCA1, considered a high risk HBOC susceptibility gene, encodes a multi-domain protein that is involved in a wide array of cellular pathways that maintain genomic stability, including cell cycle checkpoint activation, DNA damage repair, protein ubiquitination, higher chromatin hierarchical control, as well as transcriptional regulation and apoptosis [1,2,3].

Technological advances in next generation sequencing (NGS) are continuously identifying new genetic variants and many germline polymorphisms that are often difficult to interpret, named “variants of uncertain significance” (VUS). Currently, ClinVar (https://www.ncbi.nlm.nih.gov/clinvar/ accessed on 20 March 2022), one of the most used databases for variant clinical annotation, reports that about 37% of BRCA1 variants (3595) are considered VUS or difficult to interpret; this may complicate cancer risk assessment and affect the psychological state of carriers and relatives. To classify VUS and assess the clinical impact of these problematic variants, a statistical multifactorial method that considers several factors such as: tumor pathology, family history, co-segregation, and co-occurrence, was proposed [4]. However, these data may not be sufficient to achieve classification and very rare alleles are impossible to evaluate and remain VUS. Therefore, in HBOC screening, genetic methods are often not informative of functional classification owing to the low frequency of the variant and, consequently, thousands of missense variants that are identified by clinical genetic testing are annotated in public databases as VUS [5]. For these rare missense variants, functional data has emerged as a powerful way to determine whether a variant lead to loss of function. The ENIGMA consortium (https://enigmaconsortium.org/, accessed on 10 December 2021) proposed the BRCA Exchange web-tool (https://brcaexchange.org/, accessed on 20 January 2022) for sharing all the data of BRCA1/2 variants from genetic, familial, and functional studies that are reported in several databases [6]. To interpret clinical variants, the American College of Medical Genetics and Genomics and the Association for Molecular Pathology (ACMG/AMP) also recommends the use of predictive algorithms [7]. However, predictive algorithms are often non-concordant and hence, not enough to classify variants, as demonstrated recently in a study where large variation in the accuracy, particularly for benignity, was demonstrated by comparing the performance of 44 in silico prediction tools [8]. The ACMG/AMP guidelines assert that the results from “well-established” functional assays can be qualified as evidence for variant classification. In particular, ACMG/AMP proposes two sets of criteria for classifying human sequence variants that are associated with disease: one for pathogenic and one for benign variants. In both sets of criteria, functional studies are considered very useful for classification [9]. Functional data have considerable potential to aid in VUS classification and some recommendations have been recently developed for evaluating clinical reliability and validity of these assays [10,11]. Also, model organism-based functional assays are recommended to evaluate clinically-relevant variants as long as they are statistically and rigorously validated or, in the absence of rigorous statistical analysis, performed for a minimum of 11 benign and pathogenic variants [11,12]. Many assays, including those ones that are based on saturation genome editing (SGE), have been developed to assess the functional impact of BRCA1 missense variants [13,14,15]. Collecting functional data from BRCA1 Circos (https://research.nhgri.nih.gov/bic/circos/, accessed on 12 January 2022) [16], a publicly available resource, named FYI-HBOC (http://iscva.moffitt.org/fyi-hboc/build/, accessed on 12 January 2022), has been proposed to harmonize and integrate functional data for validated assays [17]. Moreover, the integration of results from a subset of validated assays (including several yeast-based assays) provided evidence for classification, according to ACMG/AMP criteria, of a total of 2355 variants [17]. This confirms the usefulness of validated functional assays in model systems such as the yeast *Saccharomyces cerevisiae,* in the functional classification of BRCA1 variants.

We previously demonstrated that the expression of BRCA1 pathogenic variants increases intra- and inter-chromosomal homologous recombination (HR) and gene reversion (GR) in two different yeast strains indicating that this phenotype could be potentially exploited as a functional evaluation of BRCA1 variants [18,19,20,21]. Here, we validated three yeast based-assays and proposed a computational approach to evaluate the clinical utility of four different BRCA1 functional assays (our assays plus an already validated yeast assay). We combined all the data by using a “classifier of majority voting” in order to give a better functional assessment of BRCA1 missense variants.

## 2. Results

### 2.1. BRCA1 Missense Variants’ Classification

The tumor suppressor gene BRCA1 encodes a multi-task protein that is mainly involved in cellular pathways controlling DNA repair and genome stability [1,2]. BRCA1 contains several domains where many missense variants have been identified and need to be studied to determine their impact on the risk of developing HBOC [4,5,6]. Since yeast is a good model system to study DNA repair proteins, we set up functional assays to get further insights on missense variants of BRCA1. We precisely selected 23 benign and 23 pathogenic variants to cover all the coding sequences of BRCA1; the benign missense variants are localized in the central part of the protein and the pathogenic variants are localized mostly in the RING and in the BRCT domain of BRCA1 (Figure 1). In addition, we have selected and analysed 10 VUS (Figure 1, Appendix A).

### 2.2. BRCA1 Pathogenic Variants Increased HR and GR in Yeast

Plasmids carrying the sequence of BRCA1 wild-type and the variants were transformed in RSY6 and RS112 strains; the transformant colonies were streaked onto glucose synthetic complete medium, lacking uracil (SC-URA), and further analyzed. To determine whether the strains were able to sustain human BRCA1 expression, total protein extracts from cells that were grown in medium containing galactose were prepared, electrophoresed, and analyzed by Western blot. The protein level of several BRCA1 variants were already evaluated previously in RS112 and RSY6 strains [18,20,21,22]. Here, we confirmed that both strains are able to express human wild-type BRCA1 and the missense variants (Appendix A). To check if the protein expression level affects the biological activity, we determined the band intensity of pathogenic and benign variants (Appendix A). By comparing band intensities, we did not observe a higher protein level of pathogenic variants with respect to the benign variants or BRCA1 wild-type suggesting that biological activity is not directly affected by the protein level.

To address whether the expression of BRCA1 variants (pathogenic, benign, or VUS) increases HR and GR, yeast cells from the diploid RS112 and RSY6 strain were incubated in galactose and plated as reported in the materials and methods (Figure 2A,B). The three assays were performed, at least, in five independent clones. 

The results were plotted as Waterfall distribution of the median values of the HR and GR frequencies. For each assay, the best cut-off and all other parameters were determined as described in the materials and methods (Figure 3A–C, Appendix A). The variants that increased HR and/or GR above the cut-off are predicted as pathogenic (Figure 3, in red); those variants that show no increase in the frequency of HR and GR and, therefore, are depicted below the best cut-off, are predicted as benign (Figure 3, blue).

The Waterfall distribution includes results that were obtained previously [18,20,21,22]. The HR and GR frequency of the strains that were transformed with the BRCA1 wild-type was used as control. The pathogenic variants that increased HR and/or GR (above the cut-off) are defined as true positive; benign variants that gave no increase in HR, and/or GR (below the cut-off) as true negative (Figure 3A–C). On the other hand, pathogenic variants gave no increase in HR and/or GR are defined as false negative, and benign variants that increased HR and/or GR as false positive (Figure 3A–C). 

By intra-chromosomal HR of the selected variants, we observed one false negative for pathogenic variants (p.R71K) and five false positives for benign variants (p.R1347G, p.I1275V, p.P1776H p.S1512I, and p.Y179C) (Figure 3A). By inter-chromosomal HR of selected variants, we observed three false negative (p.R1699W, p.A1708E, and p.R71K) pathogenic variants and two false positive benign variants (p.I1275V and p.Y179C) (Figure 3B). As far as GR assay is concerned, two pathogenic variants (p.R71K and p.R1495M) were identified as false negative, and three benign (p.I1275V, p.N132K, and p.T1675I) as false positive (Figure 3C). Remarkably, the variants p.R71K and p.I1275V were identified respectively as false negative and false positive in all three assays. 

The small colony phenotype (SCP) assay, an already validated yeast-based functional assay, is the most reliable and the best performing yeast-based test [23]. Therefore, for comparison with our assays, we performed SCP for all the selected variants and analyzed the data as reported in Appendix A. We observed that the 23 pathogenic variants scored as true positive and two benign variants (p.I1275V and p.S1512I) as false positive (Appendix A).

The performance of the yeast-based assays was evaluated and compared by plotting the receiver operating characteristic (ROC) curves and determining sensitivity, specificity, and accuracy (Table 1). Overall, the two HR assays, the GR and the SCP assay scored an accuracy value ranging from 0.870 to 0.891 to (Table 1). Among the three assays that were proposed here, intra-chromosomal HR showed the highest sensitivity (0.957) and inter-chromosomal HR the highest specificity (0.913). GR and inter-chromosomal HR recorded an accuracy of 0.891. Furthermore, from the ROC curves, we calculated the area under the curve (AUROC) and Youden Index (YI) that are used to determine the capability of each assay to differentiate between classes (pathogenic and benign). AUROC values of the three assays ranged from 0.867 (GR) to 0.929 (inter-HR) to 0.867 (GR, Table 1). By comparison, the SCP assay is more reliable than HR and GR because it recorded the highest sensitivity (1.00), accuracy (0.913), AUROC (0.951), and YI (0.913) (Table 1).

### 2.3. Effect of VUS on HR and GR in Yeast

In addition to the validation of the assays with the selected classified variants, we tested 10 BRCA1 missense variants that were classified as VUS (Appendix A). As shown in Figure 1, three VUSs (p.S592N, p.P1010S, and p.S1164I) are located in the central part of the protein, one (p.E1352K) next to SCD and six (p.A1669S, p.Y1703C, p.N1730I, p.A1789T, p.V1791I, and p.N1819S) in the BRCT domain of BRCA1. The expression of six out of ten VUSs increased intra-chromosomal HR, five induced inter-chromosomal HR, and seven VUSs increased GR (Figure 3A–C grey boxes, above the cut-off). The variants p.S1164I and p.Y1703C scored as pathogenic in the three assays; the VUSs p.E1352K, p.S592N, p.N1730I, p.V1791I, and p.P1010S scored pathogenic in two assays; p.N1819S and p.A1669S scored pathogenic in one assay; and p.A1789T scored benign in all the assays (Figure 3A–C). These observations suggest that it is quite difficult to evaluate the functional impact of these VUSs just by testing their effect on the assays that we proposed. Moreover, according to the results that were obtained with the SCP assay, the p.Y1703C, p.N1730I, p.A1789T, p.V1791I, p.E1352K, p.N1819S, and p.S592N are classified as pathogenic, whereas p.A1669S, p.P1010S, and the p.S1164 as benign (Appendix A, grey boxes). In conclusion, the results from SCP assay are not in agreement with data from HR and GR and only three out of ten VUSs gave concordant results in our three assays.

### 2.4. Development of yBRCA1: A Classifier Combination Approach for Functional Characterization

If one variant gave a positive or negative response in one specific test, it was classified as pathogenic or benign by this test (Appendix A). Precisely, 76% of the already classified variants, 35 out of 46 (23 pathogenic plus 23 benign) gave concordant response in all the tests (Appendix A). Thus, 24% of the BRCA1 variants give discordant results in all four yeast assays, making hard to give a precise indication of the functional classification. We, therefore, calculated the prediction score using the tool that was reported in http://xfer.curie.fr/get/tvsjyy4dUno/ProClass_toolbox.zip and combined all the data together by using the “classifiers combination approach” of the majority voting [24]. This classification method was named yBRCA1. By considering our tests as “classifiers”, and the “prediction scores (PS)” as votes, we assumed that a predicted score that was higher than 0.5 was suggestive of “pathogenic variant”. Consequently, variants that gave scores that were lower or equal 0.5 were classified as benign (Appendix A). In general, we can conclude that variants that gave positive or negative results in three out four yeast assays are predicted by yBRCA1 as pathogenic or benign, respectively. By this method, 2 out of 46 “prior classified” variants were not correctly classified (Appendix A). We evaluated the performance of the “combination” yBRCA1 method by calculating the specificity, sensitivity, and accuracy as reported in Table 2.

All the performance scores that are reported in Table 2 indicated that the classification method yBRCA1 is a very accurate and reliable tool for the functional classification of BRCA1 missense variants.

### 2.5. VUS Functional Characterization by the Classifier Combination Method yBRCA1 

We applied the yBRCA1 method to evaluate the functional impact and consequently the pathogenicity of 10 selected VUS. As shown in Table 3, the p.Y1703C variant is classified pathogenic in all four assays; the variants p.E1352K, p.N1730I, p.S1164I, p.S592N, and p.V1791L score pathogenic in three out of four tests. 

By combining the scores, yBRCA1 classified four VUS as benign and six as pathogenic.

We analyzed and compared the classification of the selected VUS that were obtained from yBRCA1 with that which was obtained with other approaches (Figure 4), specifically: FYI-BRCA1, RENOVO, BRCA-ML, and SGE. FYI-BRCA1 classified 2355 variants by integrating data from a set of 22 already validated functional assays [17]. RENOVO, a machine learning-based approach, has proposed a classification of 67% VUS reported in ClinVar [25]. BRCA-ML, another machine learning-based strategy, was developed to evaluate the functional impact and classify thousands of variants that were localized in the RING and BRCT domain that are reported in several databases [26]. The SGE results, made in HAP1 cell line, were collected in a database reporting the functional scores of thousands of BRCA1 variants (https://sge.gs.washington.edu/BRCA1/ accessed on 15 January 2022) [14,15].

As shown in Figure 4, out of the 10 VUS that were selected in this study, 7 were reported in FYI-BRCA1, 9 in RENOVO, 10 in BRCA-ML, and 6 VUS in the SGE database. The variant p.A1789T is classified benign only by yBRCA1 and pathogenic by the other methods. On the other hand, p.E1352K, S592N, and p.V1791L are classified pathogenic by yBRCA1 and benign by the others (Figure 4). Overall, three (p.A1669S, p.N1819S, and p.Y1703C) VUS were evaluated concordantly by all the approaches (Figure 4). This confirms that our method has the potentiality to be a promising tool to evaluate the functional impact of new VUS.

## 3. Discussion

To date, missense variants constitute the largest class of BRCA1 VUSs and, therefore, the development of new strategies to determine VUS functional classification is of pivotal importance. In clinical genetics, several predictive algorithms are used to evaluate VUS pathogenicity. Lately, the performance of SIFT and PolyPhen-2 in classifying BRCA1 and BRCA2 VUS was compared and the lack of concordance limited their application in clinics [27]. Accurate functional assays are needed to improve clinical annotation and recently, several recommendations have been proposed to report and interpret functional data [12]. Validated functional assays have largely proved to play a crucial role in the classification process [17,28]. A number of functional cell-based assays have been developed and are being used to classify BRCA1 variants [13,29]. Recently, thousands of BRCA1 variants were functionally characterized by SGE-based and complementation-based assays [14,30]. In addition, several web-tools were developed to help classify VUS. Of them, two, VarCall and FYI-HBOC, incorporate functional assay data and in silico prediction into multifactorial predictive models [17,28]. However, by these methods some variants remain VUS. The conflicting results that were observed when comparing published classification approaches suggest that it is necessary to provide further tools to increase our knowledge on the functional impact of variants. 

Yeast is a very reliable genetic system to evaluate the functional impact of disease-associated variants, and several assays, mostly based on protein biological functions, have been developed [31,32]. Here, we validated three assays, two HR assays and one GR assay, in two genetically-related yeast strains by testing 46 classified BRCA1 variants; furthermore, we tested these variants in the SCP assay that was statistically validated previously [23]. The analysis of the results of the assays one by one was not good enough to give information regarding the functional impact and the classification of the studied variants, therefore we reasoned to develop a classifier combination method, named yBRCA1, that combines and integrates the functional impact scores in all the assays that were proposed. Notably, by this method, only two variants (one pathogenic and one benign) were not correctly classified, indicating that the performance of yBRCA1 is higher than that one of one single assay.

We compared the classification that was obtained by yBRCA1, SGE, RENOVO, and BRCA-ML and observed complete concordance with the VUS classification for 3 VUS out of 10 and VUS p.A1789T was differently classified by yBRCA1 (Figure 4). Although these approaches are not directly applicable in genetic counselling and diagnostics, they provide a “functional” evaluation that could be useful to minimize the fraction of VUS and to solve conflicting interpretation. 

In conclusion, we provide a new and reliable tool for the functional categorization of BRCA1 VUS that combines validated yeast-based functional assays. This method could be very useful in the preclinical characterization of newly identified VUS because it is easy and fast to apply and showed high statistical performance. 

## 4. Materials and Methods

### 4.1. BRCA1 Variant Selection

The 23 pathogenic and 23 benign BRCA1 variants were selected according to their classification in ClinVar, BRCA Exchange, and VarSome (https://varsome.com/gene accessed on 18 January 2022) databases; the complete list of the variants is reported in Appendix A. In this study, we have also selected 10 VUS in order to validate our strategy. A total of nine VUS were identified from 3314 patients who underwent genetic counselling between 2001 and 2019 at the University Hospital of Pisa, Italy. The VUS p.A1669S was chosen because it gave conflicting results (as reported in ClinVar) and was classified as benign by a yeast-based functional evaluation [23]. Plasmids carrying the BRCA1 missense variants (Appendix A) were constructed by site-directed mutagenesis with specific oligonucleotides as reported previously [18,19,20,21]. The primer sequences that were used to construct all the plasmids carrying the variants are reported in Appendix A. All the vectors derived from YCpGAL::BRCA1 that contained BRCA1wt under the control of the galactose inducible promoter pGAL1 and URA3 marker for yeast transformant selection [33]. All the variants were checked by direct sequencing. 

### 4.2. Yeast Strains

The following strains of *Saccharomyces cerevisiae* were used: the haploid RSY6 (MAT**a** *ura3-52 leu2-3, -112 trp5-27 arg4-3 ade2-40 ilv1-92 HIS3::pRS6*), and the derivative diploid strain RS112 (*MAT**a**/MATα ura3-52/ura3-52 leu2-3,112/leu2-Δ98 trp5-27/TRP5 ade2-40/ade2-101 ilv1-92/ilv1-92 arg4-3/ARG4 his3Δ50-pRS6-his3Δ30/his3-Δ200 LYS2/lys2-801*). Complete (YPAD) and synthetic media lacking uracil (SC-URA), leucine (SC-LEU), adenine (SC-ADE), histidine (SC-HIS), or isoleucine (SC-ILE) were prepared according to the standard techniques [34].

As previously described, the diploid strain RS112 allows the measurement of intra-chromosomal HR events between two alleles *his3Δ3’* and *his3Δ5’* that are deleted at 3’ and 5’ terminus and sharing 400bp of homology; an intra-chromosomal HR event leads to the restoration of HIS3 gene allowing cells to grow in SC-HIS medium. To assess the inter-chromosomal HR, as this diploid strain carries the *ade2-40* and *ade2-101* alleles an inter-chromosomal HR event leads to the restoration of the ADE2 gene and allows cells to form colonies in SC-ADE medium [18,35]. The haploid strains RSY6 is able to measure GR events because contains the ilv1-92 allele and cannot grow in SC-ILE medium; a GR event allows cells to grow in SC-ILE [19]. Yeast strains were transformed with plasmid DNA by using the lithium acetate method with single strand DNA as a carrier, following the procedure that is described in [36]. The transformant colonies were selected in solid medium lacking uracil (SC–URA). The colonies were grown for 4 days at 30 °C and further analyzed.

### 4.3. Protein Extraction and Western Blotting

The BRCA1 protein level was determined in yeast extracts from RS112 and RSY6 strain expressing BRCA1 wild-type gene or BRCA1 missense variants. Single clones were initially grown in 10 mL of SC-URA glucose liquid medium for 24 h at 30 °C under constant shaking. Then, the cell pellet was washed twice in water and inoculated in 20 mL of SC-URA 5% galactose medium. The cells were incubated 24 h at 30 °C under shaking. Thereafter, the cultures were pelleted and washed in water. Total protein extraction was performed as previously described [19,20]. Aliquots of 10–20 μL extract were electrophoresed on a 4–10% SDS–polyacrylamide gel and transferred on nitrocellulose membrane by Trans-Blot^®^ Turbo™ Transfer System (BioRad, Hercules, CA, USA). BRCA1 was detected using Anti-BRCA1 mouse antibody that was diluted 1:300 (clone MS110, Calbiochem, Italy). As a loading control, we determined the level of the 3-Phosphoglycerate kinase (PGK) with the anti-αPGK antibody (Invitrogen, Watham, MA, USA). Western blot images were retrieved with ChemiDoc™ MP System (BioRad) and the band intensity determined by Image lab (BioRad).

### 4.4. Homologous Recombination and Gene Reversion Assays

To analyze whether the expression of BRCA1 missense variants affect intra- and inter-chromosomal HR, single colonies of RS112 strain carrying the BRCA1 expression vectors were inoculated into 5 mL of SC-URA-LEU 2% glucose medium and incubated at 30° C for 24 h (Figure 2A). Aliquots corresponding to 10^7^ cells were then incubated in 5 mL SC-URA-LEU containing 5% galactose for 17 h at 30 °C under constant shaking. Then, the cells were washed twice, counted, diluted, and appropriate numbers were plated in triplicate onto YADP to determine the number of vital cells, onto SC-HIS to evaluate intra-chromosomal HR and SC-ADE to measure inter-chromosomal HR events. The plates were incubated at 30 °C for 3–4 days until colonies were formed (Figure 2A). Frequency of intra- and inter-chromosomal HR events was expressed as number of HIS3 colonies/10^4^ total cells and the number of ADE2 colonies/10^5^ total cells, respectively [18]. 

The haploid RSY6 strain was used to evaluate the effect of BRCA1 on GR of the ilv1-92 mutation. Single colonies from RSY6 strain carrying BRCA1 vectors were grown in 5 mL SC-URA 2% glucose at 30 °C under shaking for 24 h (Figure 2B). Then, 10^7^ cells were inoculated in SC-URA medium with 5% galactose, and incubated for 17 h at 30 °C as described before. After the incubation, the cells were washed twice and the appropriate numbers were plated onto SC-ILE for scoring revertants, and in YPAD to evaluate the total cells (Figure 2B); the frequency of GR was calculated as the total number of ILV1 colonies/10^6^ cells. At least five independent experiments were carried out in strains expressing BRCA1 variants.

### 4.5. Small Colony Phenotype Assay

The expression of human BRCA1 wild-type in the budding yeast *S. cerevisiae* was found to strongly inhibit growth in solid medium. This peculiar phenotype has been exploited to develop a simple functional assay that is named small colony phenotype assay (SCP) that is based on the ability conferred by BRCA1 pathogenic variants expression to restore yeast growth [37]. For comparison, we have also analyzed all BRCA1 variants by SCP assay in the RS112 strain as follows: single colonies of RS112 carrying BRCA1 vector were inoculated in SC-URA 2% glucose for 24–48 h under shaking; then, the cells were washed and diluted. Aliquots corresponding to 150–250 cells were plated in SC-URA glucose and galactose medium. The effect on the colony size was determined by directly counting the number of cells per colony. Single colonies, picked up from each plate were resuspended in 1ml of water and the cells were counted by a hemocytometer. the results were expressed as number of cells per colony [18,37].

### 4.6. Best Cut-Off Calculation and Integration of Datasets

In this work, to evaluate the three described functional assays, we used a set of already known and classified missense variants. The results, that are expressed as the median ± standard deviation, were ordered in the Waterfall distribution and represented as box-and-whisker plots; the sensitivity, sensibility, accuracy, Youden Index (YI), receiver operating characteristic (ROC) curve, area under ROC Curve (AUROC), and best cut-off were calculated using the ProClass toolbox (http://xfer.curie.fr/get/tvsjyy4dUno/ProClass_toolbox.zip, accessed on 18 March 2022) as previously described [23,38]. BRCA1 missense variants can be divided into 3 categories on the basis of functional classification: 23 pathogenic, 23 benign, and 10 variants of uncertain significance (VUS).

The four datasets, derived from results of HR, GR, and SCP assays and obtained from classification analysis, performed by using the by ProClass toolbox, show that these assays have a general good performance (mean accuracy value of all assays is equal to 0.9 ± 0.035); therefore, a voting scheme for an ensemble-based classifier was the approach that was adopted for integrating all datasets. We applied a classifier combination method integrating all the data from all the assays which can be considered “classifiers” [24]. In the case of classification, the predictions for each label are summed and the label with the majority vote is predicted. Specifically, we used the majority voting scheme where all the responses have the same weight [24]. Simply, the variants that were classified pathogenic in 3 out of 4 assays (classifiers) gave a pathogenic prediction score (PPS) of 0.75 and were considered pathogenic. The variants that were classified pathogenic in 1 or 2 tests gave a PPS of 0.25 or 0.50 and were finally predicted as benign. 

All statistical computing and graphics were obtained by using the free software environment R, version 4.0.5, from The R Foundation for Statistical Computing (Free Software Foundation, Boston, MA, USA. 

## 5. Conclusions

Here, we have validated three functional assays in two yeast strains and developed a new method named yBRCA1 that allows to accurately establish the functional impact of new BRCA1 variants. Overall, this study also demonstrated that yeast is a reliable, fast, and not expensive system for the functional categorization of BRCA1 VUS. Importantly, yBRCA1 could be very useful to give a fast and highly accurate functional classification of newly identified VUSs.

## Figures and Tables

**Figure 1 ijms-23-04049-f001:**
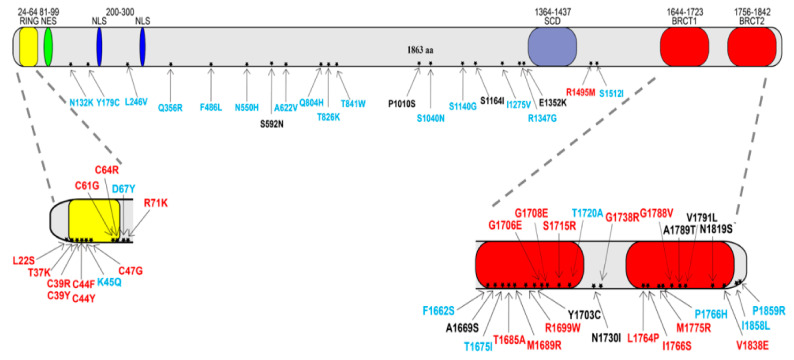
BRCA1 variant localization. BRCA1 protein structure, domains, and variant localization. BRCA1 is a protein that is composed of 1863 amino-acids. Amino-acid length is reported for any domain. RING domain, nuclear entry site, and localization signal are shown (NES, NLS); Serine cluster (SCD) and BRCT domain are also depicted. The variants, reported as amino-acid substitution, are distributed throughout all BRCA1 coding sequences; Pathogenic are shown in red, benign in blue, and VUS in black.

**Figure 2 ijms-23-04049-f002:**
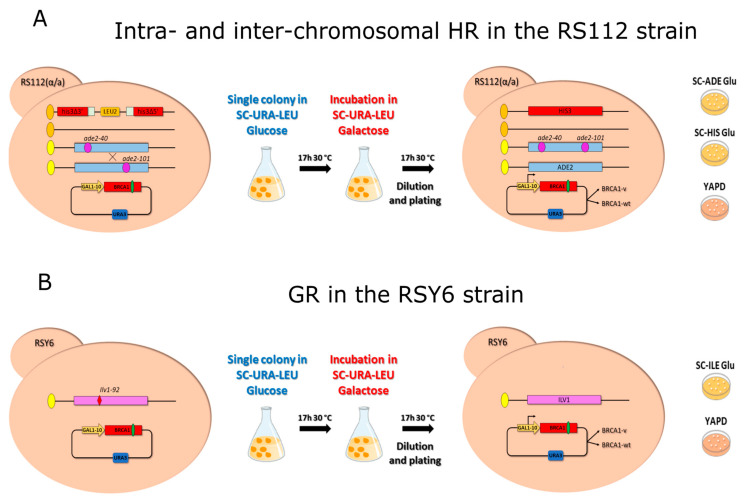
Description of the yeast-based functional assays. (**A**) HR assays were carried out in the diploid strain RS112. This strain contains the two *his3* terminally deleted alleles separated by the LEU2 marker and by the plasmid sequence, respectively, and sharing 400 bp of homology. An intra-chromosomal HR event leads to HIS3 reversion with the loss of LEU2 marker. RS112 strain also has the alleles ade2-40 and ade2-101 which allow measuring inter-chromosomal HR events. (**B**) GR assay was performed in the haploid strain RSY6 that carries the ilv1-92, by direct counting the Ilv^+^ colonies that were grown in synthetic complete medium lacking isoleucine (SC-ILE). Single colonies of the RS112 and RSY6 strain containing the plasmid-expressing BRCA1 protein or the empty vector were first pre-grown in glucose. Thereafter, 10^7^ cells were inoculated in galactose medium, as described in the materials and methods; then the cells were counted and plated to score for cell surviving fraction, intra- and inter-chromosomal HR and GR events.

**Figure 3 ijms-23-04049-f003:**
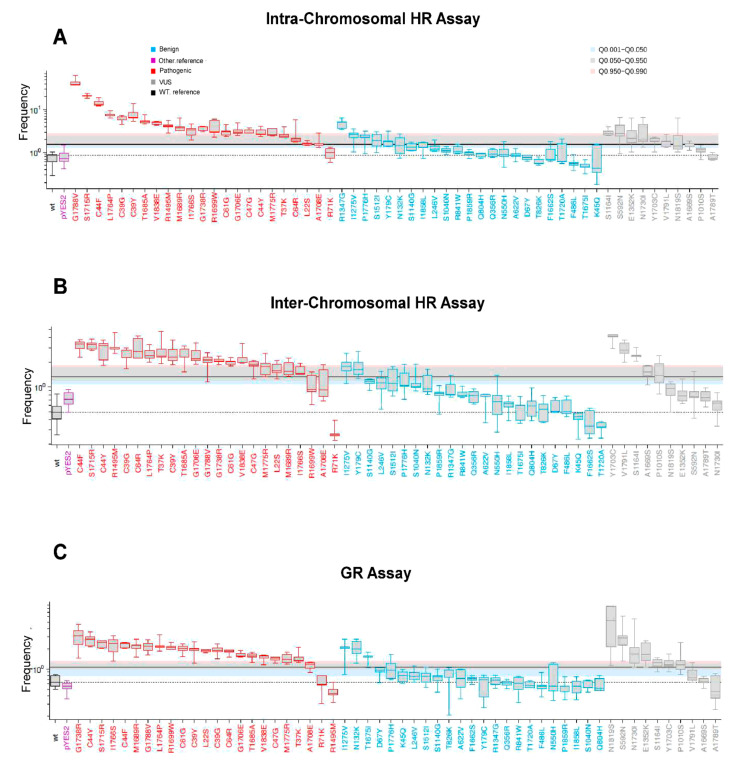
Waterfall distribution of pathogenic and benign variants and positions of best cut-offs. The data are reported as box-and-whisker plots; red, blue, and grey box indicate pathogenic, benign, and VUS, respectively. The central bar of the data box represents the median of 5–15 independent experiments; the box indicates the interquartile range (50% of data distribution), and the whiskers show the extreme values. The horizontal dotted line represents the median of control (BRCA1 wild-type). The thick line is the best cut-off; pink, grey, and light blue areas define 4%, 90%, and 4.9% of total distribution. These parameters are determined by the ProClass (http://xfer.curie.fr/get/tvsjyy4dUno/ProClass_toolbox.zip, accessed on 12 January 2022) toolbox. (**A**) Waterfall distribution of intra-chromosomal HR, (**B**) inter-chromosomal HR, and (**C**) gene reversion data.

**Figure 4 ijms-23-04049-f004:**
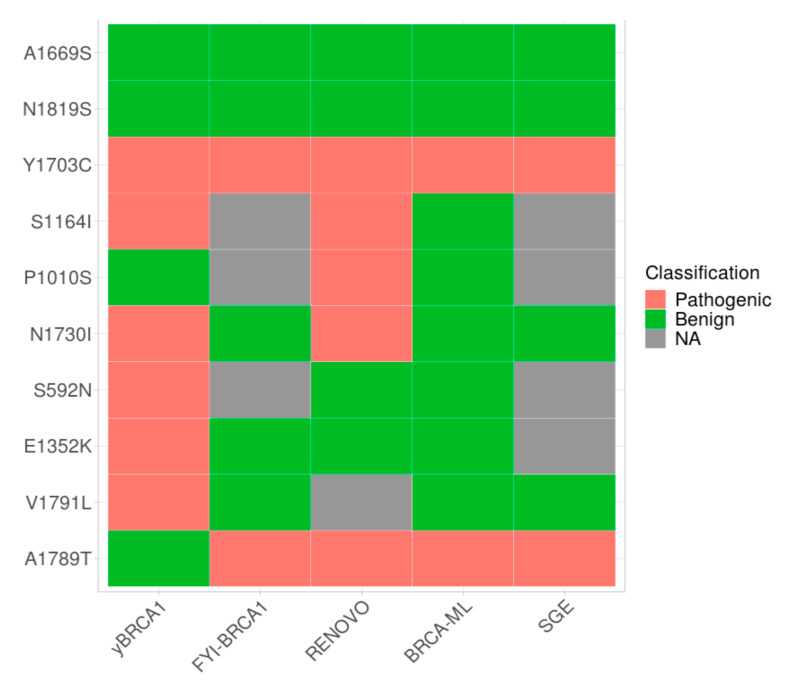
VUS classification by yBRCA1 and other methods. Heatmap showing the classification prediction of VUS that were selected in this study by yBRCA1, FYI-BRCA1, RENOVO, BRCA-ML, and SGE. NA: not applicable.

**Table 1 ijms-23-04049-t001:** Performance evaluation and reliability of yeast-based assays.

Assay	Sensitivity	Specificity	Accuracy	AUROC	YI	Best Cut-off	FP	FN
Intra-HR	0.957	0.783	0.87	0.914	0.739	1.535	Y179C I1275V R1347G S1512I P1776H	R71K
Inter-HR	0.87	0.913	0.891	0.929	0.783	1.32	Y179C I1275V	R71K R1699W A1708E
GR	0.913	0.87	0.891	0.867	0.783	1.077	N132K I1275V T1675I	R71K R1495M
SCP	1.00	0.913	0.957	0.951	0.913	25.833	I1275V S1512I	

AUROC = area under the ROC curve, YI = Youden’s Index, FP = False Positive, FN = False Negative, Sensitivity = True Positive (TP)/(TP + FN), Specificity = True Negative (TN)/(TN + FP), Accuracy = (TP + TN)/(TP + FN + TN + FP). The AUROC values were calculated using ProClass (http://xfer.curie.fr/get/tvsjyy4dUno/ProClass_toolbox.zip, accessed on 12 January 2022) toolbox and the open source R package. Sensitivity, specificity, and accuracy for intra-, inter-(chromosomal) HR, GR, and SCP assays were calculated from data that were reported in Appendix A. The best cut-off values are calculated as reported in Figure 2.

**Table 2 ijms-23-04049-t002:** Performance evaluation of the combination method “yBRCA1”.

	yBRCA1 Method
**Accuracy (CI 95%)**	0.9565 (0.8516–0.9947)
**Sensitivity**	0.9565
**Specificity**	0.9565
**AUROC**	0.9855
**Cohen’s Kappa**	0.9575
**MCC**	0.9583

Accuracy, sensitivity, specificity, and AUROC were calculated as reported in the materials and methods and in the legend of Table 1. The Matthews Correlation Coefficient is defined as:$\;MCC=\left(TPxTN-FPxFN\right)/\sqrt{\left(TP+FP\right)\left(TP+FN\right)\left(TN+FP\right)\left(TN+FN\right)}$.

**Table 3 ijms-23-04049-t003:** Pathogenic prediction of VUS by yBRCA1.

	Inter-HR		Intra-HR		GR		SCP				
Variant	PS	FI	PS	FI	PS	FI	PS	FI	PV	PPS	yBRCA1
**A1669S**	0.63575	Pathogenic	0.00675	Benign	0.00075	Benign	0.00025	Benign	1	0.25	Benign
**A1789T**	0.00025	Benign	0.00025	Benign	0.00025	Benign	0.99975	Pathogenic	1	0.25	Benign
**E1352K**	0.00025	Benign	0.56025	Pathogenic	0.99925	Pathogenic	0.99975	Pathogenic	3	0.75	Pathogenic
**N1730I**	0.00025	Benign	0.43825	Pathogenic	0.99925	Pathogenic	0.88025	Pathogenic	3	0.75	Pathogenic
**N1819S**	0.00125	Benign	0.01725	Benign	0.99975	Pathogenic	0.93575	Pathogenic	2	0.5	Benign
**P1010S**	0.40675	Pathogenic	0.00025	Benign	0.54025	Pathogenic	0.00025	Benign	2	0.5	Benign
**S1164I**	0.99975	Pathogenic	0.90575	Pathogenic	0.75425	Pathogenic	0.00025	Benign	3	0.75	Pathogenic
**S592N**	0.00025	Benign	0.89925	Pathogenic	0.99975	Pathogenic	0.99975	Pathogenic	3	0.75	Pathogenic
**V1791L**	0.99975	Pathogenic	0.30975	Pathogenic	0.00275	Benign	0.99975	Pathogenic	3	0.75	Pathogenic
**Y1703C**	0.99975	Pathogenic	0.32425	Pathogenic	0.54275	Pathogenic	0.99975	Pathogenic	4	1	Pathogenic

These variants are considered VUS according to the databases (see Appendix A for details). Predicted scores (PS) and functional impact (FI), determined by using the ProClass Toolbox (http://xfer.curie.fr/get/tvsjyy4dUno/ProClass_toolbox.zip, accessed on 12 January 2022), are reported for any assay: The inter-chromosomal HR (Inter-HR), intra-chromosomal HR (Intra-HR), gene reversion (GR), and small colony phenotype (SCP) assay. Pathogenic votes (PV) represent basically the total number of pathogenic labels; Pathogenic prediction scores (PPS) are calculated by dividing the PV by the total number of assays (4). Final prediction (yBRCA1) was evaluated by the "majority voting classification" as reported in the materials and methods.

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
