# Peer review of "Validation and Data-Integration of Yeast-Based Assays for Functional Classification of BRCA1 Missense Variants"

_ijms, 2022, doi:10.3390/ijms23074049_

Round 1
Reviewer 1 Report
Manuscript titled " yBRCA1-LR and yBRCA1-RF: two new machine learning approaches for functional classification of BRCA1 missense variants" was submitted for publication to IJMS. I would like to recommend the article for publication after considering the following comments
Line 78-83. I did not really understand why this part is included.
Line 91-95- Some details about BRCA1 would help the reader to understand the rest.
Line 12-118. Did not see any connection with the earlier part
Figures are difficult to read.
I would recommend that the titles for the table should be placed on top.
Author Response
Thanks to reviewer for the comments; the manuscript was changed according to the comments.
Line 78-83: we cancelled that part;
line 91-95: Details about BRAC1 were added;
line 112-118: Figure 1A, B, C was split into two separate figures. Figure 1A is now figure 1 and Figure1B, C is reparted as Figure 2A, B. All the figures were modifed to make them easier to read.
The title of the tables were placed on top.
Reviewer 2 Report
General remarks
The authors have addressed an important, to date unsolved question, prediction of the functional effect of missense variants, in this case of BRCA1. After establishment of three experimental assays, machine learning was applied to effectively summarise and score the assay outcome. While the manuscript is written concisely and clearly, there is a major flaw in the ML design: 10 variants of unknown signifacance (VUS) were selected as validation set, where the pathological significance is unknown, so that the ground truth and hence the accuracy of the ML models cannot be assessed properly. Apart from that, sample numbers are on the low side for ML algorithms leading to the high danger of overfitting.
Below are further remarks which might help the authors to improve their manuscript (* marks more important comments).
At the beginning of the results it would help to give an overview of the experimental design and the reasoning for it.
One general problem of the experimental setup is the variable expression levels of the proteins which most probably also affect the assay results. Therefore, it might be helpful to use different expression methods to control for that (apart from making replicate, which the authors have done).
The cross validation approach should be mentioned already in the results part, since that is crucial for the small number of samples used in this study.
Finally, in order to assemble a more suitable validation set, one would need variants not included in the test or training set and where the ground truth is known (maybe including previous VUS where the functional significance has been found later.)
Detailed remarks
line 107: explain “SC-ILE”
line 113: ‘steaked’ is probably a typo, please re-phrase
line 113: explain “SC-URA”
*line 117: Are the variable protein levels influencing the activity levels ?
line 128: “whiskers show”
line 149: false negative and false positive
line 150: explain abbreviation “SCP”
line 157: GR and SCP(?) assay
*line 164: The sentence starting with “Since AUROC values are close to 1.00...”and ending with “..are reliable at predicting the pathogenicity” should be re-phrased. 2 of 3 values are below 0.9 and “reliable” is a too strong word in this context.
line 172: should be “FP). AUROC”
*line 199: VUS as test set does not make any sense since the ground truth is not known
line 202: “normalized by dividing”
line 232: “Cohen’s Kappa”
*Fig. 1B, C: increase text font size, otherwise illegible
Fig. 3B: dashed line is missing
Fig. 4B: dashed line is missing
Author Response
Thanks to the reviewer for useful comments.
General remarks: VUS are not selected as validation set. VUS were selected just to use the ML in order to evaluate their fuctional impact. We are aware that sample numbers are low, but to avoid overfitting we have applied the and k-fold cross-validation method; therefore, the training set is randomly subdivided in k subsets alternatively used as validation and training set. This is explained at the beginning of the section 2.4. In the section 2.2 and 2.4, we have added some explaination of the experimental design and rationale of the study. We also added some detail in the matherials and methods for clarity
We have published previously several studies and expression levels were found always highly variable. As the expression is driven by the same GAL1 promoter, at RNA level ,expression should be similar. Here, we performed densitometry to compare the band intensity. As explained below, we did not observed higher expression level of pathogenic variants as compared to benign or BRCA1 wild type. Therefore, we conclude that biological activity is not directy affected by the protein level.
The cross validation approach was mentioned in the results as said before.
We have explained how the data were grouped at the beginning of section 2.4.
Detailed remarks
line 107: we explained it
line 113: we corrected into "streaked"
line113: we explained it
line 117: As said, we performed densitometry and we find that the protein level is highly variable. Moreover, by comparing the band intensity of pathogenic to neutral variants, we conclude that biological activity is not directly affected by the protein level.
line 128: corrected
line 149, 150 and 157: corrected
line 164: we cancelled the sentence
line 172: corrected
line 199: as said in the general remarks, we better explained that at the beginnings of section 2.4. In the section 2.5, we said that VUS were selected to run the MLs for the assessment of their functional impact. Some explaination has been added in the materials and methods.
line 202 and 232: corrected
Figures were all re-organized
figure 3b and 4b: we explained that in the legend of the figure and added the real values of the threshold.